# Intermittent Fasting before Laparotomy: Effects on Glucose Control and Histopathologic Findings in Diabetic Rats

**DOI:** 10.3390/nu13124519

**Published:** 2021-12-17

**Authors:** André Keng Wei Hsu, Silvane Souza Roman, Margarete Dulce Bagatini, Filomena Marafon, Paulo do Nascimento Junior, Norma Sueli Pinheiro Modolo

**Affiliations:** 1Anesthesia Department, São Paulo State University (UNESP)-Botucatu, Botucatu 18618683, Brazil; donascimentojunior@unesp.br (P.d.N.J.); norma.modolo@unesp.br (N.S.P.M.); 2Pharmacology and Histology Department, Medical School, Integrated Regional University (URI), Erechim 99709910, Brazil; roman@uricer.edu.br; 3Academic Coordination, Biomedical Sciencies Department, Federal University of Fronteira Sul, Chapecó 89802210, Brazil; margaretebagatini@yahoo.com.br; 4Postgraduate Program in Biochemistry Department, Federal University of Santa Catarina, Florianopolis 88040900, Brazil; marafon.filo@gmail.com

**Keywords:** intermittent fasting, diabetes, perioperative care

## Abstract

(1) Background: Intermittent fasting is a nutrition practice in which individuals fast for several hours in a day, mainly with feeding time during the daylight hours. They seek to improve metabolic performance and cellular resistance to stress. In this study, we tested the fasting protocol to investigate the glycemic effect in a laparotomy perioperative period in diabetic rats and histopathologic findings. (2) Methods: The animals were diabetic-induced with alloxan. Two groups were set according to the feeding protocol: free food and intermittent fasting, whose rats could only eat 8 h in the daylight. Both groups were anesthetized, and a laparotomy was performed. We evaluated the glucose levels during the perioperative period, and we accessed organ histology seeking damage of kidney, bowel and liver after surgical trauma, and we evaluated the wound healing process. (3) Results: Glycemic levels were improved in the intermittent fasting group, especially in the post-operative period after laparotomy. Comparing both groups’ tubular damage showed interdependency with mice with worse glycemic level (Z = 2.3; *p* = 0.0215) and wound-healing parameters showed interdependency with rats with better glycemic status for neovascularization (Z = 2.2; *p* = 0.0273) and the presence of sebaceous and sweat gland in the healing process (Z = 2.30; *p* = 0.0215). (4) Conclusions: Intermittent fasting before surgery can be a tool to improve glycemic levels in diabetic rats, with improvement especially in the post-operative period.

## 1. Introduction

Intermittent fasting (IF) is a nutrition strategy that is showing promising results in metabolic improvement in glucose regulation, weight loss and stress resistance [1]. In this practice the hours of the feeding day are restricted, alternating periods from 12 to 24 h of fasting in some days of the week. This aims to promote cellular resistance, organism resilience and improve plasticity when damage occurs [2].

In animal models, prophylactic IF diminished progression and gravity in several conditions such as cardiovascular diseases (myocardial infarction and stroke) [3,4], cancer and improvement of cancer-related therapy (chemotherapy or immunotherapy) [5,6].

In a diabetes scenario, previous studies have shown that IF in murine models promotes better β cell function and decreases insulin resistance [7,8]. In addition, gut microbiome is restructured and prevents retinopathy and prolonged survival [9]. In humans, results are attributed to weight loss in association with enhancement of peripheral insulin action [10,11].

In surgical field, this nutritional strategy could be a new method to improve outcomes and prepare the organism for the trauma of the surgical process. In animal models of ischemia/reperfusion injury, IF has displayed protection in hepatic and renal ischemic damage and has reduced intimal hyperplasia [12,13,14].

Considering that diabetes is a very challenging condition in the perioperative period and uncontrolled blood glucose can lead to worse surgical outcomes [15], we hypothesized that IF before surgery could be a new tool to improve glucose metabolic endurance and control after surgical trauma in diabetic rats, since the preoperative period represents an opportunity to optimize glycemic levels and potentially decrease adverse events [16].

## 2. Materials and Methods

The study was conducted according to ARRIVE guidelines and under the approval and regulations set forth by Institutional Animal Care and Use Committee of Universidade Regional Integrada do Alto Uruguai—Erechim/Ceua (resolution 1628/CUN) on 19 March 2019 according to Brazilian animal studies law [17]. Wistar rats (all 3 months of age males *n* = 40) were housed in appropriate pathogen-free controlled conditions at a temperature of 20–24 °C in a group of four individuals in each cage. Light cycle was fixed at 12 h (12 h light and 12 h dark, starting at 8:00 am). All rats received water ad libitum and standard food according to the fasting protocol.

Experimental diabetes was induced in 12 h fasted rats by single intraperitoneal injection of alloxan [18] (150 mg/kg body weight) dissolved in citrate buffer (100 mM, pH 4.5). To prevent fatal hypoglycemia, as a result of massive pancreatic insulin release, the rats were supplied with 10% glucose solution after 6 h of alloxan administration for the next 24 h [18]. After seven days, rats with glycemia ≥200 mg/dL were selected for the experiment (*n* = 13).

The rats were divided in two feeding groups: the ad libitum (free food—FF) and the intermittent fasting protocol. In both groups the number of daily calories and nutrients were the same: 20 g per rat daily. In the FF group the amount of food was divided in two and was given at 8:00 a.m and the other half at 20:00 p.m. In the IF protocol, they fasted for 16 h and were fed for 8 h a day. The fasting hours were based on previous studies [19]. All amount of food was placed in the beginning of the light cycle at 8:00 a.m to mimic food intake only in the daylight and was taken at 16:00 p.m. They interspersed two days of fasting to one of free food where the food was given as the same as the FF group. They had 15 days of this feeding cycle before the laparotomy surgery.

All rats were anesthetized with 80 mg/kg ketamine + 10 mg/kg xylazine intraperitoneally in the right lower abdominal quadrant. For multimodal analgesia metamizole 150 mg/kg, 2 mg/kg morphine were administered subcutaneously, and the operative wound was infiltrated with 0.5% 2 mg/kg lidocaine. Under aseptic conditions a 3 cm laparotomy was performed, and abdominal wall closing was performed with continuous 2.0 polypropylene stiches and 3.0 nylon for the skin. After the surgery, both groups returned to FF diet protocol.

During the feeding protocol, blood glucose was measured (One Touch Ultra Mini^TM^/Johnson & Johnson do Brazil, Complexo JK, São Paulo, Brazil) in both groups in the mornings after the fasting days to search for possible hypoglycemia, and immediately after the surgery every 6 h.

In 48 h after the surgery, rats were euthanized with thiopental 150 mg/kg and lidocaine 10 mg/kg intraperitonially. Kidney, liver, the duodenal intestinal segment and the surgical wound were collected for histological assessment (experiment is diagrammed in Figure 1).

For histological analysis, the material was preserved, placed in paraffin and slides stained with hematoxylin and eosin were performed. A histologist analyzed and qualitatively classified the lesions into absent, mild, moderate and severe (0, 1, 2, 3). Kidney evaluation involved the degree of glomerular and tubular damage, for the liver, the presence of steatosis and necrosis. For duodenal segment we evaluated villi destruction and presence of inflammatory cells. Finally, in the operative wound we checked for the presence of inflammatory cells, re-epithelialization, fibroplasia, collagenization, vasodilatation, neovascularization and the presence of sebaceous and sweat glands in the healing process.

### Statical Analyzes

All statistical analyses were performed by a data scientist in the R software with the RStudio (Version 4.1.0 (22 June 2020), RStudio, Inc. Boston, MA, USA). The sample size was calculated using data from a previous study that was independent of the research reported herein [20].

Mixed linear models were conducted to investigate changes in intergroup glucose level (IF vs. FF) at each time point and intragroup across the pre-laparotomy fasting protocols and post-laparotomy evaluations. Furthermore, we also used this model for changes in weight and waist circumference of rats over time. The residual error of all mixed linear models showed a distribution assumed to be Gaussian by quantile–quantile plots and by the Shapiro–Wilk test. For the pre-laparotomy assessments of glucose and waist circumference, the Box–Cox transformation was used. The adjusted mixed linear models included the interaction between groups and moments as a fixed effect and rats as a random individual effect. Multiple comparisons were performed in the post-test with the correction of the Bonferroni procedure, assuming a significance of 5%.

To identify interdependencies between groups (IF or FF), glucose ranges and qualitative histopathology findings, a multiple correspondence analysis (MCA) was conducted. A priori, the glucose values collected 48 h post-laparostomy were categorized into 4 classes according to extreme values (minimum and maximum) as well as the first, second, third and fourth quartile, namely: from 284 to 300, from 301 to 374, from 375 to 412 and from 413 to 483 mg/dL. A posteriori, the matrix formed by the groups, classes of glucose and the qualitative findings of the histopathology was converted into a Burt table and submitted to the chi-square test to investigate the existence of a non-random relationship between the rows and columns of the matrix, as well as to extract the residuals (observed value subtracted from the expected value) that were adjusted and standardized by the normal z-score. Therefore, the residuals standardized by the z-score (Z) >1.65 and >1.96 determined the existence of interdependencies with significance of, respectively, 10 and 5% between the classes of qualitative variables; the greater the distance from the point of cut established by Z, the greater the magnitude of interdependence. When Z was <−1.65 or <−1.96, it indicated the non-existence of interdependencies with significance of, respectively, 10 and 5%.

Finally, tests were shown in a two-dimensional perceptual map. The interdependencies of interest was pointed out by the Z, as well as the construction of confident ellipses with the density distribution of rats from both experimental groups for a qualitative visual judgment. To confirm the separation between the ellipses of the groups, the individual score of each rat was extracted from the first two dimensions, which is a coefficient representing the multiple interdependence between the variables. Variations in individual scores as a function of groups were investigated by simple linear regression. The Shapiro–Wilk test was performed, and multiple comparisons were performed with the correction of the Bonferroni procedure using 5% significance.

## 3. Results

Between intragroup comparisons throughout the pre-laparotomy fasting protocols, rats in the IF group showed higher glucose in the first moment compared to the fourth and there was a balance in the variation of glucose from the fifth, while for those in the FF group the glucose in the third moment was lower compared to the sixth, tenth and eleventh, representing an instability in the glycemic curve over the 11 moments. In the intergroup comparison, rats from the IF group had lower glucose compared to those from the FF group in the second, fourth, sixth, eighth, ninth, tenth and eleventh fasting moments. Glucose changes suggest a lower and more stable glycemia in the IF group rats throughout the 11 fasting protocols (Figure 2A). Furthermore, rats in both groups reduced their weight after the 11 fasting protocols and only those in the IF group showed a reduction in waist circumference (Figure 2B). In both groups the amount of food intake was the same, 20 g/rat/daily, and there were no food leftovers. Due to this non-variation, no analyses were conducted for this item.

In the post-operative glucose control, rats in the IF group exhibited higher glucose at 0 h after surgery compared to 30, 36, 42 and 48 h post-laparotomy and there was a stabilization of glycemia variation from 6 h after the operation. In the FF group, they also showed higher glucose at 0 h after the procedure compared to 30, 36 and 42 h post-laparotomy, however after 6 h the post-operative glycemic curve remained unstable (Figure 2C). In the intergroup comparison, rats in the IF group had lower glucose compared to those in the FF group at all evaluation times. Such changes in glucose suggest lower and more stable glycemia in rats in the IF group after laparotomy. The MCA showed that rats with glucose > 413 mg/dL (fourth quartile) regardless of group had interdependence with tubular damage score 3 (Z = 2.3; *p* = 0.0215). Animals with glucose from 301 to 374 mg/dL (second quartile) showed interdependence with score 3 for neovascularization (Z = 2.2; *p* = 0.0273) and sebaceous and sweat gland (Z = 2.30; *p* = 0.0215). The other histological parameters did not show statistical interdependence. Furthermore, the distance between the ellipses on the perceptual map suggests a separation of animals submitted to intermittent fasting from those with free feeding (Figure 3). The visual separation of the ellipses plus the Z findings suggests a response pattern in which rats with better glycemic control showed milder damage to the renal tissue, a greater degree of neovascularization and the presence of sebaceous and sweat glands healing processes after laparotomy. Histological images are shown in Figure 4.

## 4. Discussion

In this study, we demonstrated that IF before surgery remarkably reduced the levels of blood glucose in diabetic rats in the perioperative period, mainly with better stabilization of the glycemic curve in the post-operative period comparing with the FF group. It is also important to say that the animals did not have any important hypoglycemia during the fasting protocol in the IF group.

Our findings in this surgical field are in agreement with numerous previous studies demonstrating that IF is a beneficial intervention in diabetes. The mechanisms shown in the preceding research could explain our perioperative glucose results: IF improves insulin sensitivity with less deterioration of pancreatic islets and reconstructed gut microbiota with bacteria correlated with better glucose control [21,22,23]. Since surgery is an aggression to homeostasis and could bring difficulty to glucose control, IF could be a protection against surgical trauma since it increases cellular stress resistance [24] and improves metabolic flexibility [25]. In diabetic-induced rats in the previous analysis, IF improved glucose tolerance, increased plasma insulin and improved *β*-cell mass [26]. The adoptive stress response of IF includes prevention of inflammation, better handling of oxidative stress and formation of more mitochondria [27].

In addition to all the metabolic benefits of IF, especially in glucose metabolism, studies have shown secondary advantages that could be interesting for the physicians in the perioperative period. In pain, IF enhanced morphine-induced antinociception while mitigating tolerance and constipation in mice [28]. Another potential benefit is the cognitive function that could ameliorate against distress [29]. In general body weight in intermittently fasted rats is usually lower than their ad libitum fed counterparts, but in our study both groups lost weight and the difference was not statistically significant. However only in the IF group there was a reduction in abdominal circumference (*p* < 0.05) [30].

In our histological findings we investigated the IF effect on wound healing, kidney, liver and duodenal bowel damage after surgical trauma. In our results, shown on the perceptual map, less renal impairment was close to the ellipse of IF with rats with better glucose control compared with higher levels, with significant interdependency of moderate tubular damage on the third quartile of glucose, represented by the FF group. The same occurred with wound-healing parameters with better healing status close to IF ellipse comparing with FF. Furthermore, there was significant interdependency of neovascularization and presence of sebaceous and sweet glands on the wound-healing process. It is difficult to say that these results are due to IF or to the better glucose control promoted by the fasting; however, both are very intertwined. It is clear that individuals with better glucose control have better healing outcomes [31]. We conjecture that the results from both have synergy, since during in vitro studies fasting enhanced endothelial angiogenesis brought activation of SMOC1 and SCG2, facilitating neovascularization and wound healing [32].

Renal impairment is always worrisome in the perioperative period. Our diabetic kidney finding after laparotomy followed the same path of previous studies where IF protected against tubular damage in induced ischemia–reperfusion injury. This points out that IF could be a nutrition protection promoting renal endurance against injury after surgical trauma [33,34]. In our study it is not clear if this result is because only of IF or the better glucose control was promoted by it, or both.

Another concern of the researchers was the increase in liver steatosis in the perioperative period. Fortunately, there was no interdependency between groups, with no increase in steatosis or liver damage in both groups. It is important to point out that in previous studies IF promoted reduction of steatosis and inflammation in high fed fat or fructose mice [35]. Unfortunately, our bowel results did not show important interdependence in villi destruction and inflammation.

Our study has several limitations. First, we did not investigate if the mechanisms behind our findings were the same that IF promoted in other conditions in experimental studies. We speculate that IF in the perioperative period targets the same multiple pathophysiological mechanisms of protection as shown in ischemia–reperfusion injuries studies, such as preservation of mitochondrial homeostasis, conservation of redox status, suppressing endoplasmic reticulum stress and abrogated signaling pathways (ERK1/2 and mTOR) [36]. Another potential limitation is that we did not investigate the functionality of the organs, we only assessed the morphology status to compare and elucidate histological findings in possible damage and the effect in the surgical wound. Furthermore, our most important limitation is that we could not dissociate whether the histological results are due to IF or to the better glucose promoted by it, or both. Hence, perioperative organ injury is always a concern to perioperative physicians, and there is no specific pharmacological therapy proven effective in the prevention of the damage [37]. This study of IF before laparotomy could guide new studies to confirm the mechanisms and benefits of IF in the surgical field.

## 5. Conclusions

In this study, we demonstrated that glucose control in diabetic rats is ameliorated in the perioperative period with intermittent fasting, with less renal impairment and better surgical wound healing. Human testing will be required in further studies since IF may be a new resource to improve organism endurance to surgical stress, especially in diabetic subjects.

## Figures and Tables

**Figure 1 nutrients-13-04519-f001:**
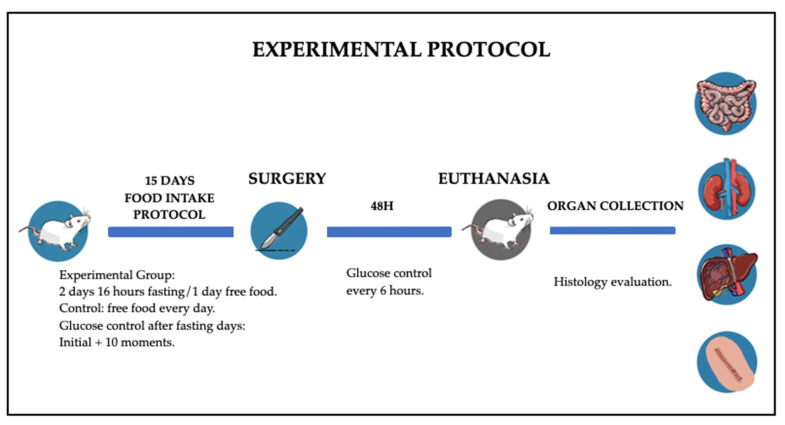
Schematic image of experimental protocol in diabetic rats of our study.

**Figure 2 nutrients-13-04519-f002:**
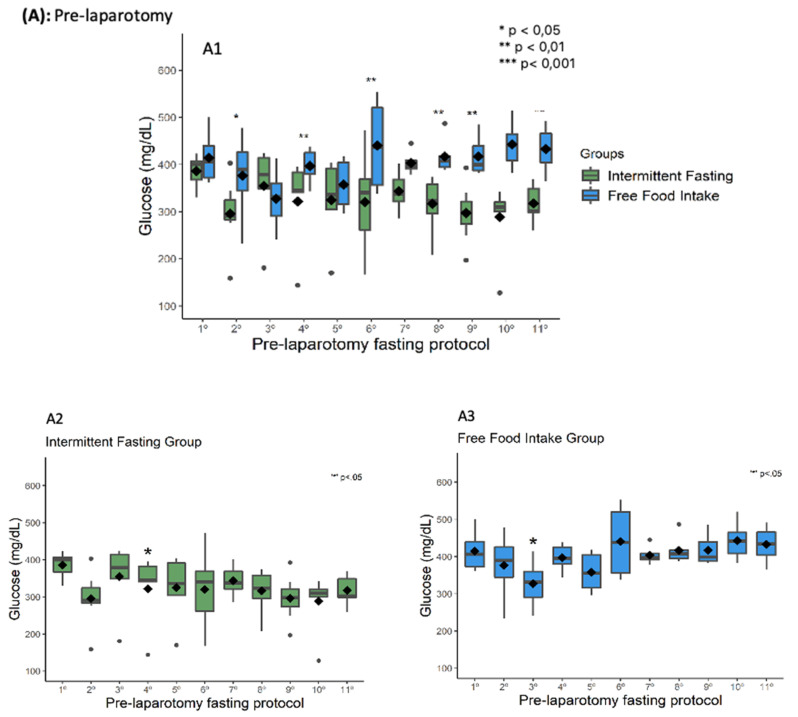
Green represents IF group. Blue represents FF group. * *p* < 0.05 ** *p* < 0.01 *** *p* < 0.001. The lower and upper bounds of the box represent the first and third quartiles of the data, respectively; the horizontal line plus space inside the box indicates the median; the diamond indicates the mean; black circles indicate outliers; comparisons were performed by mixed linear models. (**A**) Pre-laparotomy fasting protocol glucose levels showing comparison between groups (**A1**) and comparison over time in relation to the first time-point for only intermittent fasting group (**A2**) and free food intake group (**A3**). (**B**) Comparison of the weight and abdominal circumference between groups (**B1**/**B2**) and comparison over time in relation to the starting time-point for only intermittent fasting group (**B3/B4**) and free food intake group (**B5**/**B6**). (**C**) Post-laparotomy fasting protocol glucose levels showing comparison between groups (**C1**) and comparison over time in relation to the first time-point for only the intermittent fasting group (**C2**) and free food intake group (**C3**).

**Figure 3 nutrients-13-04519-f003:**
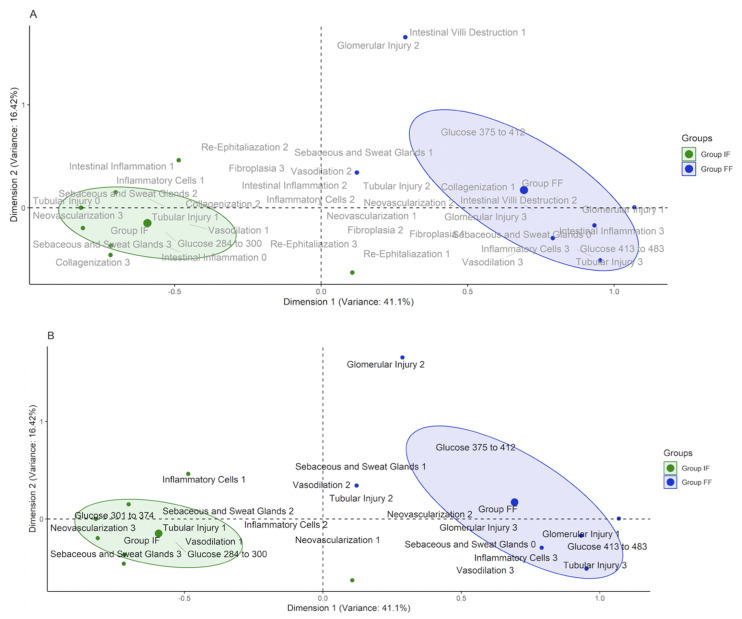
Perceptual map of the first and second dimensions of the multiple correspondence analysis with the qualitative findings of histopathology, experimental groups and glucose classes. In (**A**), all the qualitative variables applied in the analysis are presented, and in figure (**B**) only those with interdependencies by the chi-square test.

**Figure 4 nutrients-13-04519-f004:**
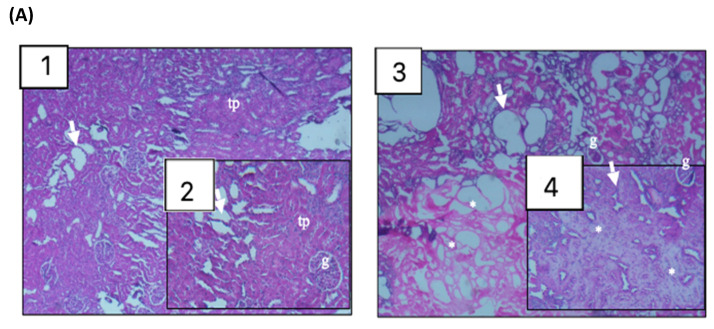
Box 1;2 represents IF group while Box 3;4 represents FF group. (**A**)—Kidney: tp: proximal tube; g: glomerulus; arrows: tubular damage; *: proximal and distal tubular damage. (**B**)—Liver: vc: centrilobular vein; arrows: hepatocytes; *: necrosis. (**C**)—Duodenal segment: arrows: intestinal villi; *: inflammatory cells. (**D**)—Operative Wound: E: epithelium; p: hair follicles.

## Data Availability

The data presented in this study are available on request from the corresponding author.

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
