# Peer review of "Intermittent Fasting before Laparotomy: Effects on Glucose Control and Histopathologic Findings in Diabetic Rats"

_nutrients, 2021, doi:10.3390/nu13124519_

Round 1

Reviewer 1 Report

great research

Author Response

Thank you very much for the review. 

Reviewer 2 Report

 Hsu et al. investigated the effects of intermittent fasting on surgical stress, including hyperglycemia, and wound healing process using diabetic model rats and found that intermittent fasting prior to laparotomy suppressed blood glucose levels in post-operative period. A multiple correspondence analysis showed interdependence between renal tubular damage and higher glucose levels and that between wound healing parameters and lower glucose levels. They conclude that intermittent fasting before surgery can be a novel preventative strategy to achieve less renal injury and better wound healing after surgery, especially in diabetic patients. Their approach is very interesting and valuable since postoperative complications are a great issue that still has no definitive pharmacological strategy. However, the reviewer considers that this manuscript has some important concerns before acceptation for publication in Nutrients. Details are listed below.

1) To make Figure 2, the authors investigated histological alteration in some organs and tissues, such as kidney, liver, bowls, and observed wound healing parameters, such as neovascularization and presence of sebaceous and sweat Gland. The authors should add new figures which includes the comparisons between intermittent fasting (IF) and free food (FF) groups and the representative images.

2) As the authors mentioned in Discussion, the present study lacks investigation into mechanisms by which intermittent fasting improved blood glucose levels, histological injury, and wound healing process. At least, any speculations and citations will be needed.

3) “In both groups the number of daily calories and nutrients were the same, 20g per rat daily” in Materials and Methods (Page2 Line69-70): Were there any differences in amount of food intake between IF and FF groups? It is better to show the food intake data in Results.

4) How did the authors decide the fasting protocol? Is there their own preliminary or others’ previous studies?

In addition, the explanation is not clear. Were there 11 fasting protocols (Figure 1A) in 15 days of the feeding cycle (Page2 Line74-76)?

Author Response

Response to Reviewer 2 Comments:

Dear reviewer, thank you for your time and suggestions to improve our manuscript. We adjusted the problems that you noticed in our paper and the answers are pointed below (all changes are marked in yellow in text to facilitate the review):

Point 1: To make Figure 2, the authors investigated histological alteration in some organs and tissues, such as kidney, liver, bowls, and observed wound healing parameters, such as neovascularization and presence of sebaceous and sweat Gland. The authors should add new figures which includes the comparisons between intermittent fasting (IF) and free food (FF) groups and the representative images.

Response 1: We included figure 3 with histological images between the groups to visually enrich Figure 2 and make more representative our results for the reader. Page7 Line200

Point 2:  As the authors mentioned in Discussion, the present study lacks investigation into mechanisms by which intermittent fasting improved blood glucose levels, histological injury, and wound healing process. At least, any speculations and citations will be needed.

Response 2: We included our speculations and new citations about probably mechanisms behind the study results. Blood glucose levels: Page9 Line 232-235. Histological Injury: Page10 Line 276-282; citation 35.  Wound healing process: Page9 Line260-263; citation 31.

Point 3: “In both groups the number of daily calories and nutrients were the same, 20g per rat daily” in Materials and Methods (Page2 Line69-70): Were there any differences in amount of food intake between IF and FF groups? It is better to show the food intake data in Results.

Response 3: We included the food intake data in the results. Page4 Line152-153

Point 4:  How did the authors decide the fasting protocol? Is there their own preliminary or others’ previous studies?

In addition, the explanation is not clear. Were there 11 fasting protocols (Figure 1A) in 15 days of the feeding cycle (Page2 Line74-76)?

Response 4: We decide the fasting hours for the protocol based on previous studies. Page 2 Line 74-75; citation 19. We had 11 fasting protocols because we measured glucose only on the mornings after fasting days to search for possible hypoglycemia, to make cleaner we explicated better in the text (Page3 Line 85-87). Thus, we had the first measure plus 2 days of fasting in the protocol 5 times (1+ 10).

We hope we have made all the necessary adjustments in the manuscript to suit Nutrients Journal.

Thank you for the review and the affection with our study.

André Hsu.

Round 2

Reviewer 2 Report

The reviewer still has the same concerns before acceptation for publication in Nutrients.

1) Not only representative photos but also quantification should be needed for comparison between IF and FF to show the protective effects of intermittent fasting.

2) Speculation into mechanisms should be discussed more logically with more citations

3) The authors should add the data of the following description: “There were no differences in amount of food intake between IF and FF groups” (Page4 Line153-154).

Author Response

Response to Reviewer 2 Comments: 

Dear reviewer, thank you for your time and suggestions to improve our manuscript. We adjusted the problems that the academic editor asked by email. The answers to your questions are pointed below (all changes asked by academic editor are marked in yellow in text to facilitate the review):

Point 1: Not only representative photos but also quantification should be needed for comparison between IF and FF to show the protective effects of intermittent fasting.

Response 1: In the materials and methods we described the histology evaluation methodology (absent, mild, moderate, and severe) by the histologist Page 3 Line 93-99. Since this is a qualitative data, is difficult to quantify in numbers in the figures. This is the reason that we proposed a multiple correspondence analysis and the perceptual map to represent qualitative data. 

Point 2: Speculation into mechanisms should be discussed more logically with more citations

Response 2: In the discussion we had already communicated that this is one of ours study limitation. We also included in the conclusion that furthers studies will be necessary to evaluate the mechanisms and human tests are necessary Page13 Line 331-336. Some speculations and new citations about probably mechanisms behind pre-laparotomy IF and glucose control were done, but we prefer not to theorize too much without confirmation. Since this is an unprecedent study and in perioperative area we do not have yet any proven pharmacology therapy to prevent surgical damage we figure out that our study could guide new ones. In anesthesiology field, fortunately, there is a global trend to enhance surgical results with better outcomes. However, in current guidelines (like ERAS – Enhanced Recovery After Surgery) there are no significant evidence yet of which diet days before surgery can enhance surgical outcomes, so our study could be one step to caught perioperative researchers attention to this important theme. 

Point 3: The authors should add the data of the following description: “There were no differences in amount of food intake between IF and FF groups” (Page4 Line153-154).

Response 3: We included the food intake data in the results: 20g/rat/daily in both groups, with no food leftovers. Due to this non-variation, no analyzes were conducted for this item.  Page 5 Line 158-160.

We hope we have made all the necessary adjustments in the manuscript to suit Nutrients Journal. 
Thank you for the review and the affection with our study.
André Hsu. 
